# Investigation of the Superposition Effect of Oil Vapor Leakage and Diffusion from External Floating-Roof Tanks Using CFD Numerical Simulations and Wind-Tunnel Experiments

**Jie Fang, Weiqiu Huang \*, Fengyu Huang, Lipei Fu and Gao Zhang**

Jiangsu Key Laboratory of Oil & Gas Storage and Transportation Technology, Changzhou University, Changzhou 213164, China; fangxyjoyce@sina.com (J.F.); 17000271@smail.cczu.edu.cn (F.H.); fulipeiupc@163.com (L.F.); e9940610@Gmail.com (G.Z.)

**\*** Correspondence: hwq213@cczu.edu.cn

**Abstract:** Based on computational fluid dynamics (CFD) and Realizable k-ε turbulence model, we established a numerical simulation method for wind and vapor-concentration fields of various external floating-roof tanks (EFRTs) (single, two, and four) and verified its feasibility using wind-tunnel experiments. Subsequently, we analysed superposition effects of wind speed and concentration fields for different types of EFRTs. The results show that high concentrations of vapor are found near the rim gap of the floating deck and above the floating deck surface. At different ambient wind speeds, interference between tanks is different. When the ambient wind speed is greater than 2 m/s, vapor concentration in leeward area of the rear tank is greater than that between two tanks, which makes it easy to reach explosion limit. It is suggested that more monitoring should be conducted near the bottom area of the rear tank and upper area on the left of the floating deck. Superposition in a downwind direction from the EFRTs becomes more obvious with an increase in the number of EFRTs; vapor superposition occurs behind two leeward tanks after leakage from four large EFRTs. Considering safety, environmental protection, and personnel health, appropriate measures should be taken at these positions for timely monitoring, and control.

**Keywords:** external floating-roof tank; oil vapor superposition effect; numerical simulation; leakage and diffusion; wind tunnel

## 1. Introduction

External floating-roof tanks (EFRTs) are widely used for crude oil storage [1]. With the development of petroleum reserve strategies, different types of EFRTs have been developed. However, the floating deck in an EFRT cannot seal a tank wall absolutely as it needs to float up and down freely [2]. In other words, there is an annular rim gap between the floating deck and tank wall. As the elasticity of the rim seal gradually decreases with long-term usage, the rim gap widens. Especially, improper operation or poor maintenance will aggravate the attrition of the sealing device. Under such conditions, oil evaporation from the rim gap and oil vapor diffusion into the atmosphere increase simultaneously. The discharged vapor typically contains volatile organic compounds (VOCs), which can cause several safety and environmental problems [3–5]. When air flows through storage tanks, vortices are generated and an improper layout will produce some 'dead angles of the vortices' [6], and these dead angles adversely affect air flow and oil-vapor discharge. In this case, the concentration of oil vapor in these dead angles is superimposed, which increases the likelihood of accidents, such as fire. Therefore,

analysing the effect of superposition on VOC leakage and diffusion in EFRTs has obvious practical significance and theoretical value [7–9].

Numerical simulation methods are widely used to describe oil-vapor diffusion in storage tanks [10–12]. Sharma et al. [13] investigated static breathing evaporation loss from two horizontal storage tanks on the ground and underground and found that higher the concentration of n-butane and i-pentane, higher is the breathing loss. Huang et al. [14] and Wang et al. [15] investigated the effects of oil loading rate and the initial oil-vapor concentration on the oil-vapor mass transfer and the evaporation loss in the large doom roof tank by using the phase-interface convection mass transfer model. The results revealed the variation rules of the oil-vapor concentration, the speed ratio of gas to liquid, and the evaporation loss rates of oil products in the tank and at the discharge ports. Hou et al. [16] investigated the heat and mass transfer mechanisms in refueling process by using two-dimensional unsteady state model of the vehicle refueling process. The results showed that as the refueling velocity increases, the gas-liquid mixing is increased, and the free surface of liquid is gradually blurred. Hassanvand et al. [17,18] used the volume-of-fluid (VOF) model of CFD to simulate the various influence factors in the process of gasoline tank loading, and studied the effects of the temperature, the oil loading speed, the initial oil-vapor concentration of the tank on the oil loss rate of the tank. Hao et al. [19] carried out numerical simulation methods and experimental verification for the oil vapor leakage and diffusion from the large and small EFRT at different leakage locations and pore sizes. The results showed that when there is a rim leakage between the floating deck and tank wall, oil vapor diffuses along the tank wall to the upper space of the floating deck. Ai and Mak [20] used CFD methods under the hypothesis that infectious respiratory aerosols exhausted from a unit can reenter into another unit in the same building through opened windows, and found that the distribution of the polluted gas is highly dependent on the wind direction, and the diffusion is more intense when the wind deviation angle is not $0^0$.

Several researchers used the wind-tunnel test platform to study oil leakage and diffusion from storage tanks [21–23]. Liu et al. [24] studied the diffusion behaviour of heavy gases in the case of instantaneous leakage and continuous release in wind tunnels. Using this methodology, the influence of different obstacles on the diffusion of heavy gases was also studied. Macdonald et al. [25] used the wind-tunnel test platform to study wind loads on tank walls and roofs of different types, tank sizes, and Reynolds numbers. Poterla and Godoy [26] carried out experimental studies on cylindrical shells with different height-diameter ratios and roof forms in a wind tunnel and obtained the corresponding wind-pressure distribution law. Wang et al. [27] measured the volume fraction of carbon dioxide, ethyne and propylene in a flammable gas-leak accident on direct-current wind-tunnel test platform, analysed the concentration distribution using a meteorological chromatograph, and measured the wind speed distribution using an anemometer. A range of hazardous gase volume fraction was obtained at different wind speeds and different leakage rates.

The diffusion of oil vapors is highly dependent on the ambient wind speed. At different wind speeds, vapor distribution trends in a tank vary, resulting in different concentration distributions and vapor-accumulation locations. Furthermore, there may appear superposition effects of wind speed and concentration fields in different EFRT groups. Therefore, in this study, we conducted wind-tunnel experiments and numerical simulations on a single EFRT and two EFRTs at different ambient wind speeds of 2, 4, and 6 m/s. Subsequently, numerical simulations were conducted on vapor leakage and diffusion from four 10000 m$^3$ EFRTs.

## 2. Methodology

### 2.1. Experimental Protocol

A self-made direct-flow wind tunnel was used to generate steady wind fields, as shown in Figure 1. The wind tunnel (DFWT-10) included gas-gathering, stable, contraction, test, first diffusion, power, and second diffusion sections. The size of the test section is 1.5 m (H) × 1.5 m (W) × 3 m (L) and the

turbulence intensity of the designed wind field in the test section is 30–40% to simulate a wind field (0–20 m·s$^{-1}$). The ambient wind speed, the temperature, and the humidity can be measured by the hot-wire anemometer (TES-1341, Taishi, the wind speed range of 0–30 m·s$^{-1}$ and the resolution of 0.01 m·s$^{-1}$, the temperature range of −10–60 °C and the resolution of 0.01 °C, and the humidity range of 10–95% RH and the resolution of 0.1% RH). The evaporation loss can be automatically measured by the high-precision electronic balance (WT-30000-1B, Wantai Electronic Balance with the range of 0 - 30 kg and the resolution of 0.1 g). The mass difference method was used to measure the mass change of n-hexane in a period of time. N-hexane mass was measured before and after the experiment and the mass change can be calculated as the mass loss of n-hexane during an hour, which is measured for 5 times. Then, the variation of the mass per unit time can also be calculated as the loss rate of n-hexane. In addition, the evaporation loss rate of the EFRT from the annular rim gap were measured by the wind tunnel test, and then the evaporation loss rate was set as the mass-flow-inlet of the boundary conditions of the annular rim gap in the FLUENT software. The gas sampler (QC-4S) with a rate range of 0.1–1.5 L·min$^{-1}$ was chosen to sample the vapor around the tank. The vapor components and concentrations can be analysed using a gas chromatography (GC-2010 Plus, Shimadzu International Trading Co., Limited, Japan) with FID and capillary column of Rtx-1 (30 m × 0.25 mm × 0.25 μm).

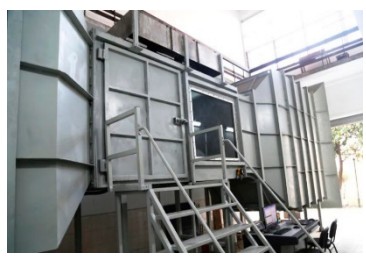 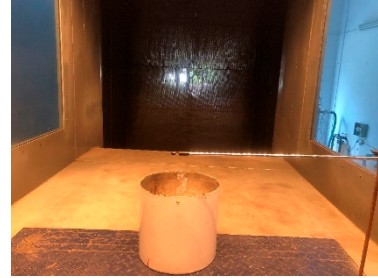 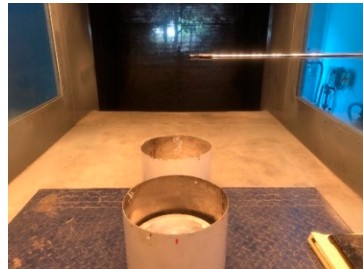

**Figure 1.** Wind tunnel for the experiments.

The small EFRT represents a scaled model (35:1) of a 1000 m$^3$ field EFRT [28]. This ratio guarantees the blocking rate of the tank in the wind tunnel. The diameter, wall height, and rim gap width of the small EFRT were 344, 272, and 6 mm, respectively. A schematic diagram of the wind-tunnel experiment is shown in Figure 2. Because the volatility of n-hexane is moderate, it was used as a representative of conventional oil in the numerical calculation of the leakage and diffusion from EFRTs. When the wind speed is 2 m/s, *Re* for the field in the wind tunnel is 257566, which is more than 4000, so it can be seen as turbulence.

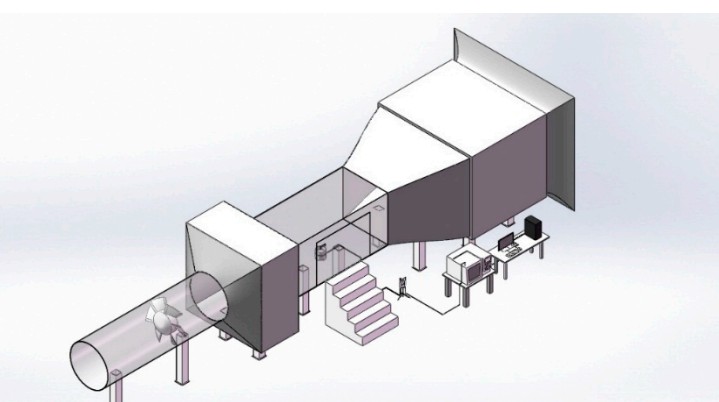

**Figure 2.** *Cont.*

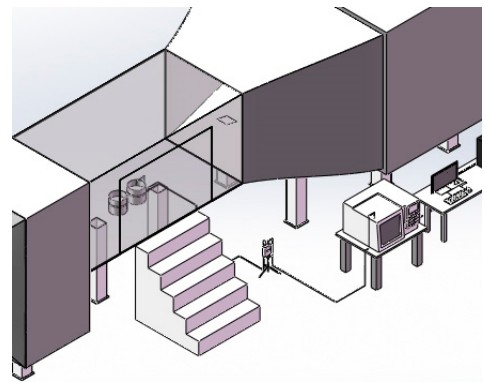

**Figure 2.** Schematic representation of the wind-tunnel experiment.

*2.2. Numerical Calculation Method*

2.2.1. Governing Equations

An EFRT is affected by external wind and the gas space above the floating deck and around the tank wall will produce a pressure difference. Due to this pressure difference, oil vapor under the seal rim of the floating deck will diffuse into the atmosphere. To describe this fluid motion, the following governing equations and turbulence model were used.

(1)   The continuity equation,

$$div(u) = \frac{\partial \rho}{\partial t} + \frac{\partial}{\partial x_j}(\rho u_j) = 0 \tag{1}$$

where $\rho$ (kg·m$^{-3}$) is the fluid density, $t$ (s) is the time, $x_j$ (m) represent the moving distance on X, Y, and Z axes, and $u_j$ (m·s$^{-1}$) represents velocity vectors on X, Y, and Z axes. For the incompressible fluid, the density is the constant.

(2)   The momentum equation,

$$\frac{\partial(\rho u_i)}{\partial t} + \frac{\partial}{\partial x_j}(\rho u_i u_j) = -\frac{\partial p}{\partial x_i} + \frac{\partial}{\partial x_j}(\mu_t \frac{\partial u_i}{\partial x_j}) + (\rho - \rho_a)g_i \tag{2}$$

where $p$ (Pa) is the absolute pressure of the atmosphere, $\mu_t$ (Pa·s) is the eddy viscosity, $\rho_a$ (kg·m$^{-3}$) is the density of the atmosphere, and $g$ represents gravitational acceleration. The subscript i in $x_i$, $u_i$ and $g_i$ indicates the values on X, Y, and Z axes, respectively.

(3)   The energy equation,

$$\frac{\partial(\rho E)}{\partial t} + \frac{\partial(\rho u_j E)}{\partial x_j} = \rho f_j u_j - \frac{\partial(p u_j)}{\partial x_j} + \frac{\partial(\tau_{ij} u_j)}{\partial x_i} + \frac{\partial}{\partial x_j}\left(k\frac{\partial T}{\partial x_j}\right) + S_h \tag{3}$$

Here,

$$E = h - \frac{p}{\rho} + \frac{u^2}{2}$$

where $T$ (K) is the temperature of the fluid, $f_j$ (N/(m$^{-3}$·s)) is the volume force, $\tau$ is the stress tensor, $S_h$ includes the heat of the chemical reaction, and any other volumetric heat sources.

(4)　The component transport equation,

$$\frac{\partial(\rho\omega)}{\partial t} + \frac{\partial}{\partial x_j}(\rho u_j \omega) = \frac{\partial}{\partial x_j}\left(\rho D_l \frac{\partial \omega}{\partial x_j}\right) \tag{4}$$

Here,

$$\omega = \frac{CM_{mol}}{1000\rho} = \frac{nM_{mol}}{1000\rho V}$$

where $D_1$ (m²/s) is the turbulent diffusion coefficient. $\omega$ is the mass fraction of the vapor to the gas mixture of the vapor-air. $C$ (mol/L) is the molar concentration of the vapor, $M_{mol}$ (g/mol) is the molar mass of the vapor, $n$ (mol) is the amount of the vapor, $V$ (m³) is the volume of the vapor-air mixture. This equation is applied to systems with mass exchange or multiple chemical components.

(5)　The turbulence model

In general, an EFRT is located in the atmospheric boundary layer above the ground in industrial applications. Flow field in the boundary layer is affected by air pressure, temperature, ground friction, obstacles, and other parameters and hence, the flow is turbulent. Both the standard $k$-$\varepsilon$ model and realizable $k$-$\varepsilon$ turbulence model can be employed to simulate fully-developed turbulent flow; however, the latter better represents flow separation and vortexes than the former; furthermore, the realizable $k$-$\varepsilon$ turbulence model yields a more accurate concentration distribution than the RNG $k$-$\varepsilon$ turbulence model [29]. Thus, the realizable $k$-$\varepsilon$ turbulence model was chosen for numerical calculations; the turbulent kinetic energy and dissipation rate equations of the model are shown in Equations (5) and (6), respectively.

$$\frac{\partial(\rho K)}{\partial t} + \frac{\partial(\rho u_y K)}{\partial x_y} = \frac{\partial}{\partial x_y}\left[\left(\mu + \frac{\mu_t}{\sigma_K}\right)\frac{\partial K}{\partial x_y}\right] + P_K + G_b - \rho\varepsilon - Y_M \tag{5}$$

$$\frac{\partial(\rho\varepsilon)}{\partial t} + \frac{\partial(\rho u_y \varepsilon)}{\partial x_y} = \frac{\partial}{\partial x_y}\left[\left(\mu + \frac{\mu_t}{\sigma_\varepsilon}\right)\frac{\partial\varepsilon}{\partial x_y}\right] + \rho C_1 S\varepsilon - C_2\rho\frac{\varepsilon^2}{K + \sqrt{v\varepsilon}} + C_{\varepsilon 1}\frac{\varepsilon}{K}C_{\varepsilon 3}G_b \tag{6}$$

Here,

$$\mu_t = C_\mu\rho\frac{K^2}{\varepsilon},\ C_{\varepsilon 1} = 1.44, C_2 = 1.9, \sigma_\varepsilon = 1.2, \sigma_K = 1.0,\ C_1 = \max\left(0.43, \frac{\eta}{\eta + 5}\right)$$

$$\eta = \frac{K}{\varepsilon}S,\ S = \sqrt{2S_{xy}S_{xy}},\ C_\mu = \frac{1}{A_0 + A_s\frac{U^*K}{\varepsilon}}$$

$$A_0 = 4.04,\ A_S = \sqrt{6}\cos\varphi,\ \varphi = \frac{1}{3}\arccos\left(\sqrt{6}W\right),\ W = \frac{S_{xy}S_{yz}S_{zy}}{\sqrt{S_{xy}S_{xy}}},\ S_{ij} = \frac{1}{2}\left(\frac{\partial u_x}{\partial x_y} + \frac{\partial u_y}{\partial x_x}\right)$$

$$U^* = \sqrt{S_{xy}S_{xy} + \widetilde{\Omega}_{xy}\widetilde{\Omega}_{xy}},\ \widetilde{\Omega}_{xy} = \Omega_{xy} - 2\varepsilon_{xyz}\omega_z,\ \Omega_{xy} = \overline{R}_{xy} - \varepsilon_{xyz}\omega_z,\ \overline{R}_{xy} = \frac{1}{2}\left(\frac{\partial u_x}{\partial x_y} - \frac{\partial u_y}{\partial x_x}\right)$$

In these equations, $\rho$ (kg·m⁻³) represents fluid density, $f_x$ (N·m⁻³) represents volume force, $\mu$ (Pa·s) is the kinetic viscosity, $K$ (m²·s⁻²) is the turbulent kinetic energy, $\varepsilon$ (m²·s⁻³) is the dissipation rate, $P_k$ (m·s⁻²) is the turbulent kinetic energy generation term, $G_b$ is the buoyancy generation term, $Y_M$ is the compressibility corrected term, $v$ (m²·s⁻¹) indicates kinematic viscosity, and $\omega_z$ (rad·s⁻¹) indicates angular velocity. When the direction of shear flow is the same as the gravitational direction, $C_{\varepsilon 3} = 1$ and when the shear flow is perpendicular to the direction of gravity, $C_{\varepsilon 3} = 0$. $\sigma_k$ and $\sigma_\varepsilon$ are the Prandtl

numbers corresponding to the turbulent kinetic energy and dissipation rate, respectively; $S_k$ and $S_\varepsilon$ are user-defined values.

### 2.2.2. Computational Domain and Boundary Conditions

The computational domain size setting should take into account both the calculation time and the accuracy of calculation results. In computational wind engineering, the blocking ratio is often used to set the cross-sectional area of the computational domain. If the blocking ratio is less than 3% to 5%, it is considered that the flow field near and in the tank is not affected by the boundaries of the computational domain [30]. Considering the computational accuracy, blocking ratio, and calculation time, as shown in Figure 3, a three-dimensional computational domain was selected in this study. The size of the region was 15D (X) × 5H (Y) × 10D (Z) (D: tank diameter, H: total height of the tank). Figure 3a shows the computational domain of a single small EFRT and Figure 3b shows the domain corresponding to two small EFRTs. Large EFRTs are commonly used in industrial applications; as shown as Figure 3c, four 10000 m$^3$ EFRTs were chosen to investigate the effect of oil vapor superposition between EFRTs. Structured mesh division was selected. The total number of cells for the single, double and four EFRTs was about 1.10 million, 1.74 million and 2.44 million, and the independence of cells were validated.

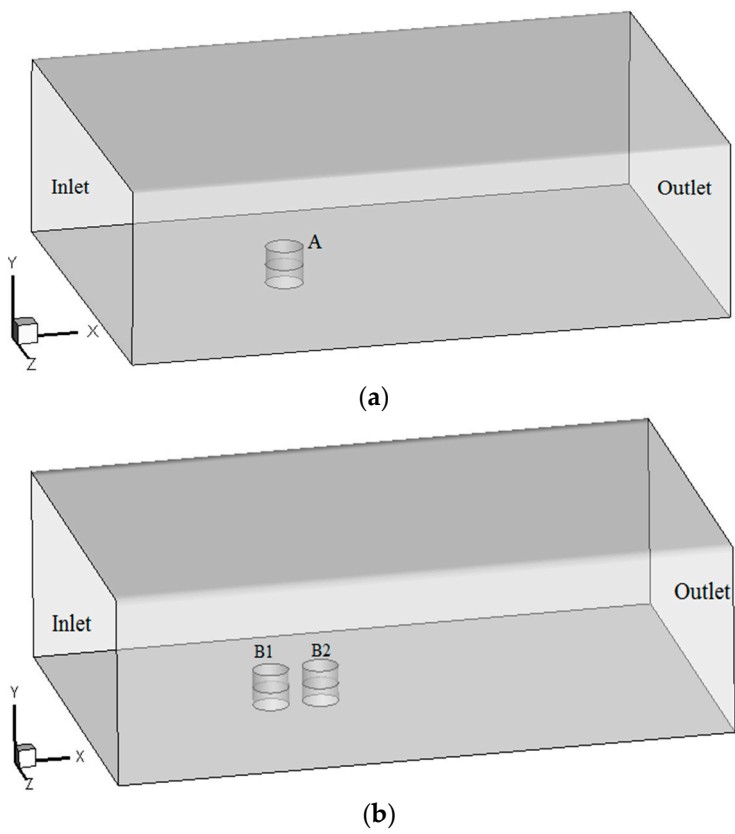

(a)

(b)

**Figure 3.** *Cont.*

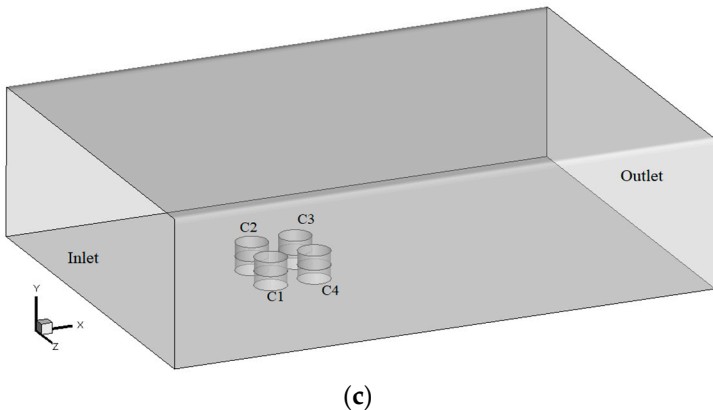

(**c**)

**Figure 3.** Computational domain of (**a**) a single external floating-roof tank (EFRT), (**b**) two EFRTs, and (**c**) four large EFRTs.

The inlet boundary of the flow field was set as the velocity inlet boundary condition and the ambient wind speed represents an exponential distribution. Wind speed was introduced using the FLUENT User Defined Function (UDF). The direction of wind speed was positive along the X axis. The outlet boundary of the flow field was set as the pressure outlet boundary condition while the gap between the floating deck and the tank wall was set as the mass-flow boundary condition and mass-flow rates were determined experimentally. The tank bottom, tank wall, and floating deck were all set as no-slip boundaries and the ambient temperature was set at 13.5 °C. The mass-flow rate of the single tank at 2 m/s is $2.37 \times 10^{-5}$ kg·s$^{-1}$, at 4 m/s is $4.30 \times 10^{-5}$ kg·s$^{-1}$ and at 6 m/s is $5.13 \times 10^{-5}$ kg·s$^{-1}$. The mass-flow rates of the double tanks at 2 m/s are $3.52 \times 10^{-5}$ kg·s$^{-1}$ (B1) and $2.42 \times 10^{-5}$ kg·s$^{-1}$ (B2). The mass-flow rates of the double tanks at 4 m/s are $5.08 \times 10^{-5}$ kg·s$^{-1}$ (B1) and $4.30 \times 10^{-5}$ kg·s$^{-1}$ (B2). The mass-flow rates of the double tanks at 6 m/s are $6.25 \times 10^{-5}$ kg·s$^{-1}$ (B1) and $5.03 \times 10^{-5}$ kg·s$^{-1}$ (B2). The parameter properties in the calculation process are shown in Table 1.

**Table 1.** The properties of material parameters in the calculation process.

| Material | Test Temperature/°C | Density/kg·m$^{-3}$ | Mole Mass/g·mol$^{-1}$ | Saturated Vapor Pressure/kPa | Diffusion Coefficient in Air/$10^{-6}$ m$^2$·s$^{-1}$ |
|---|---|---|---|---|---|
| n-hexane vapor | 13.5 | 663.5 | 86.2 | 11.9 | 7.4 |
| atmosphere | 13.5 | 1.29 | 29 | / | / |

## 3. The Wind-Tunnel Test Validation

There are many factors affecting leakage and diffusion from EFRTs, including the position of the floating deck, ambient wind speed, and temperature. At present, there are few experimental studies on the leakage and diffusion laws of EFRTs. To understand evaporation loss from EFRTs at different ambient wind speeds (2, 4, and 6 m/s) and oil vapor distribution inside or outside EFRTs and to verify the rationality of the simulation and EFRT geometric models applied to oil-evaporation loss, a wind-tunnel test platform was used for experimental research and data analysis under leaking conditions in the rim gaps of the floating decks of EFRTs. Herein, the floating deck height was defined as the distance of the floating deck position to the tank bottom and it was set at 136 mm. Since n-hexane is the main component of gasoline vapor, and its physical properties are relatively mild, it is feasible and convenient to use n-hexane instead of gasoline for experiment and simulation.

Firstly, the height of the floating deck was set at 136 mm, i.e., the space below this height was filled with n-hexane. Later, the ambient wind speed was varied from 2 to 6 m/s. The wind speed and concentration-field distributions at the same position of the floating deck height but at different ambient wind speeds were measured. The measuring positions for the single EFRT were located at the centre of the single EFRT (A) (W1), 0.2D behind the single EFRT (A) (W2), 0.9D behind the single EFRT

(A) (W3), and 1.6D behind the single EFRT (A) (W4). The measuring point positions of the two EFRTs were the centre of the windward EFRT (B1) (P1), 0.2D behind the windward EFRT (B1) (P2), centre of the leeward EFRT (B2) (P3), and 0.2D behind the leeward EFRT (B2) (P4). Herein, the position of W3 corresponded to that of P3 and the position of W4 corresponded to that of P4. In these stated values, D represents tank diameter.

From the above experiments, the wind- and concentration-field distributions at different ambient wind speeds were obtained, as shown in Figures 4–6. In these figures, as gas chromatographic measurements were calibrated using methane, the values of concentration fields were based on methane concentration.

According to Figures 4–6, the larger the ambient wind speed, the greater is the disturbance from the leeward EFRT (B2) to the windward EFRT (B1), which is mainly reflected in the larger the maximum wind speed above the windward EFRT (B1) than that above the leeward EFRT (B2). The concentration distribution at 2 m/s is different from that at 4 and 6 m/s. Vapor concentration above the windward EFRT (B1) is lower than that above the centre of the two EFRTs at a wind speed of 2 m/s, which shows that most of the vapor is still in B1; the vortex current above the centre of the two EFRTs leads to a higher vapor concentration than that just above the windward tank (B1).

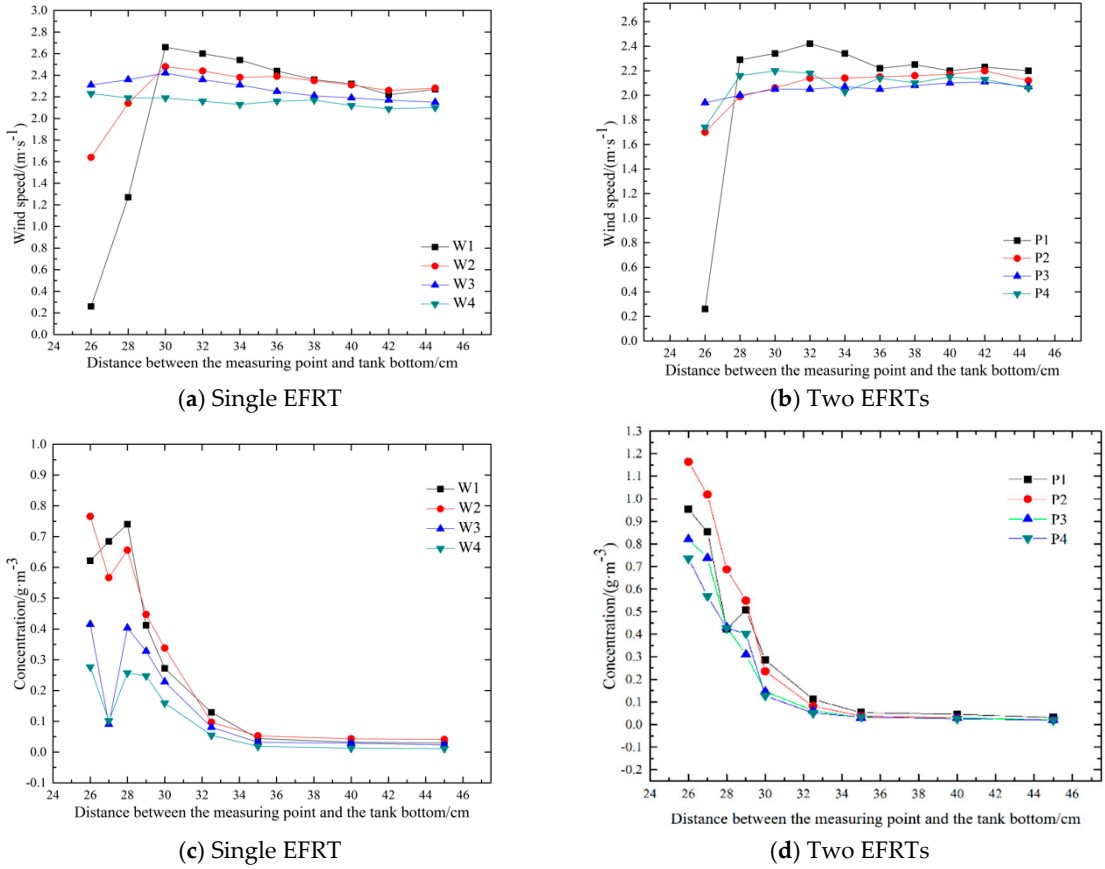

**Figure 4.** Wind speed distribution and concentration distribution above the floating deck surface at an ambient wind speed of 2 m/s.

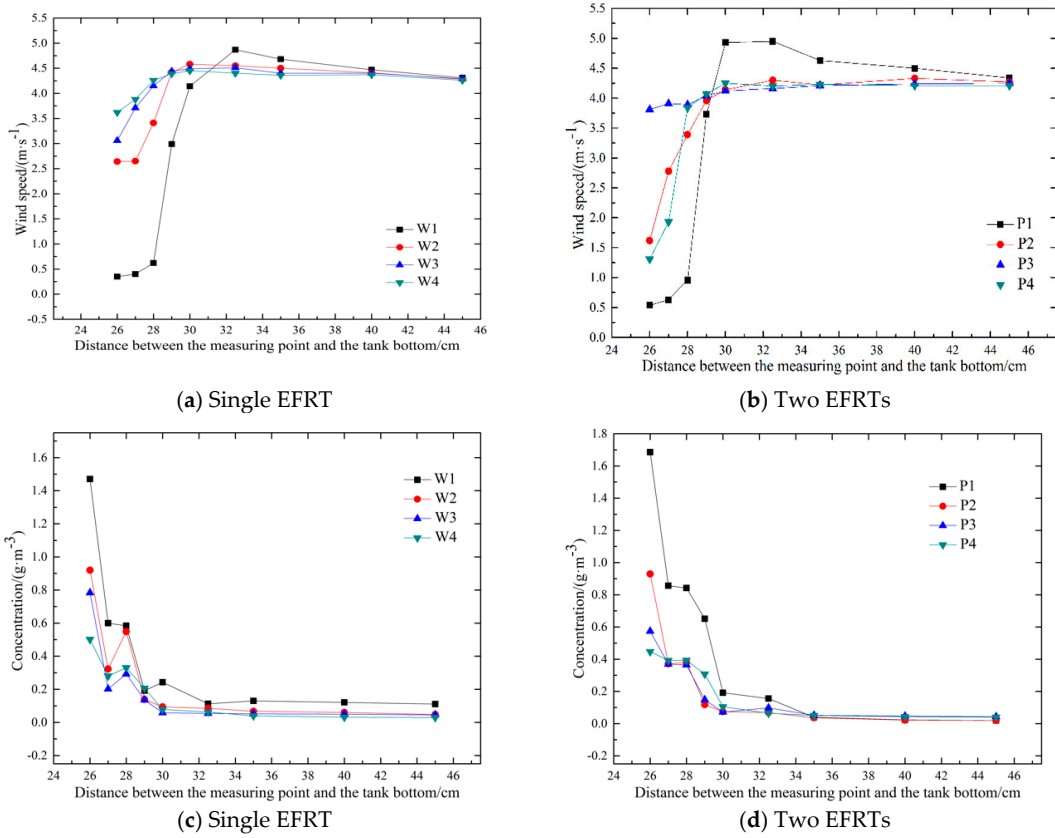

**Figure 5.** Wind speed distribution and concentration distribution above the floating deck surface at an ambient wind speed of 4 m/s.

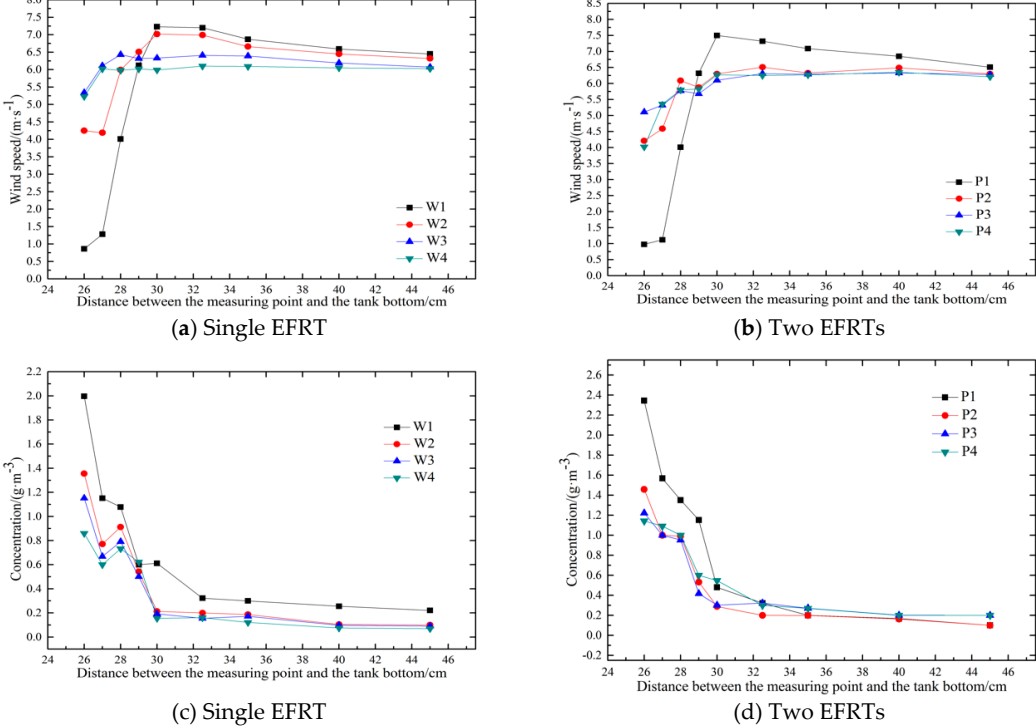

**Figure 6.** Wind speed distribution and concentration distribution above the floating deck surface at an ambient wind speed of 6 m/s.

At an ambient wind speed of 2 m/s, the values of wind speed and vapor concentration at each point in the vertical wind direction above the centre of the floating deck of the single EFRT (A) and above the centre of the floating decks of the two EFRTs (B1 and B2) were measured and they were then compared with the simulated values. The results are shown in Figures 7 and 8. From these figures, it can be inferred that the simulated values are consistent with the experimental values with only a small error between them, which proves that the construction of the geometric model and settings used for the numerical calculation method are reasonable. The deviations in wind speed are mainly due to errors in measurement. The probe of an anemometer affects the flow field to a certain extent when it enters into the tank. The deviation in concentration is mainly due to the destruction of some concentration fields around the sampler when it extracts vapor. Improper cleaning of the sampler also affects the measurement results.

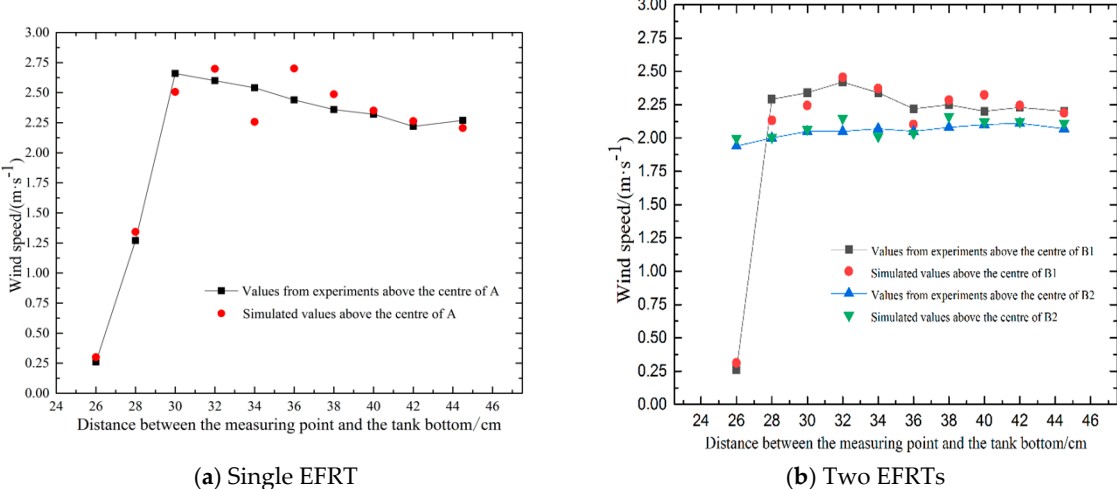

(**a**) Single EFRT　　　　　　　　　　　　　　　(**b**) Two EFRTs

**Figure 7.** Comparison between experimental and simulated wind speed distribution values.

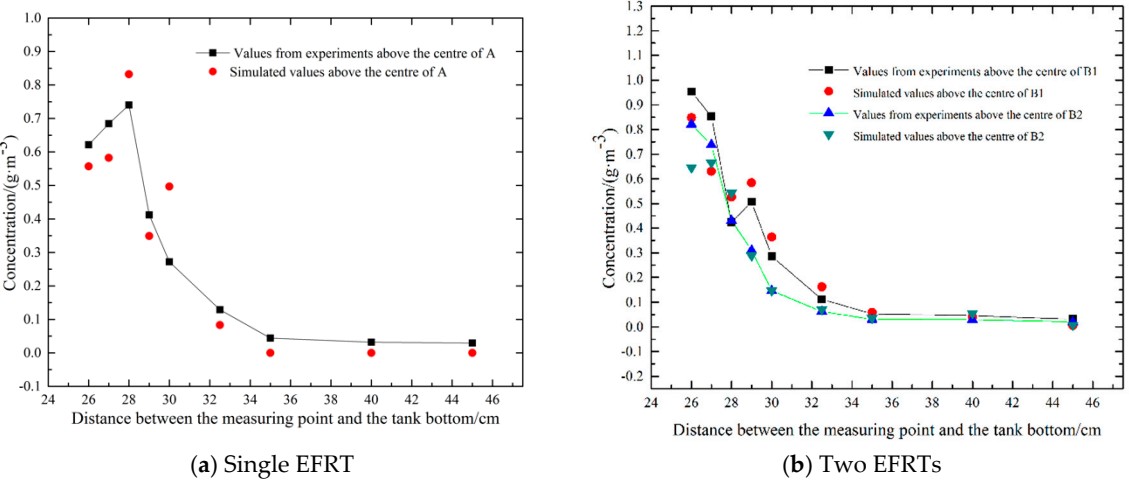

(**a**) Single EFRT　　　　　　　　　　　　　　　(**b**) Two EFRTs

**Figure 8.** Comparison between experimental and simulated vapor-concentration distribution values.

## 4. Results and Analysis

### 4.1. The Wind Speed Distribution of Different EFRTs

Based on CFD numerical computations, the wind speed distributions of various EFRTs (single, two, and four) were analysed at a floating deck height of 122 mm. Wind speed cloud diagrams of the single EFRT and two EFRTs on the XY plane along the X-axis were simulated (Figure 9). To conveniently compare velocity distributions at different ambient wind speeds, Figure 9a,d represent

the values obtained at 2 m/s. Figure 9b,e show the values corresponding to a wind speed of 4 m/s while Figure 9c,f show the values corresponding to a wind speed of 6 m/s.

Wind speed cloud diagrams of the single EFRT on the XY plane along the X axis were simulated (Figure 9a–c). It can be seen in the figures that irrespective of the ambient wind speed, the following phenomena occur. On the windward side of the tank, due to blocking, airflow speed decreases gradually to 0 m/s; there is a light blue area close to the tank wall due to the reverse airflow caused by wind hitting the tank wall. At the bottom area of the windward side, there is a blue area with a negative wind speed, indicating backflow in this area and the danger of vapor superposition. On the leeward side of the tank, a large blue area with a negative wind speed appears on the right side of the tank, which indicates that the leeward area of the tank has a strong backflow and the wind speed isopleth is not as close to the tank wall as that on the windward side and the entire airflow-speed isopleth inclines along the lower right side. In the area above the tank, there is a high airflow-speed area (red area), where the wind speed exceeds the ambient wind speed. This is because the airflow area above the tank is smaller, leading to an accelerated airflow rate.

Wind speed cloud diagrams of the two EFRTs on the XY plane along the X axis were simulated (Figure 9d–f). It can be noted in these figures that wind speed distribution on the windward side is basically similar to that of the single EFRT. Although the airflow-speed values are different in the back area, the entire airflow-speed isopleth inclines along the upper right side. The space of the blue backflow zone at the back becomes larger and more complex. This is due to mutual blocking between tanks, which aggravates turbulence. The above-described phenomena occur irrespective of the ambient wind speed.

Combining with the wind fields measured experimentally, it can be found that the wind speed at 2 m/s is slightly different from that at 4 and 6 m/s. When the ambient wind speed is 2 m/s, the maximum wind speed above B1 is lesser than that above A but at 4 and 6 m/s, the maximum wind speed above B1 is approximately similar to or larger than that above A.

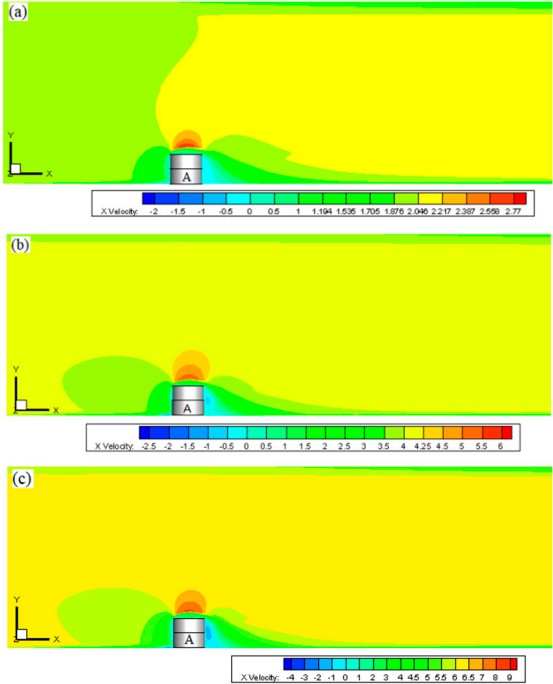

**Figure 9.** *Cont.*

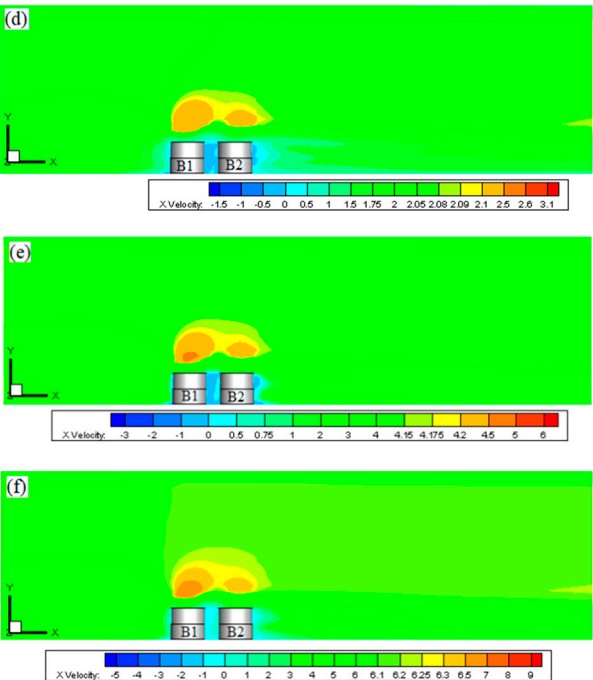

**Figure 9.** Wind speed cloud diagrams of a single EFRT and two EFRTs on the XY plane along the X axis at ambient wind speeds of (**a**,**d**) 2, (**b**,**e**) 4, and (**c**,**f**) 6 m/s.

The wind speed cloud diagrams of the four large EFRTs on the XY plane along the X axis were simulated (Figures 10 and 11) at an ambient wind speed of 4 m/s. Figure 10 shows the wind speed cloud diagrams of C1 and C4 and Figure 11 shows the wind speed cloud diagrams of C2 and C3. According to these figures, wind speed distribution on the windward side is similar to that observed in the case of the single EFRT and two EFRTs. The rule of area between C1 and C4 (C2 and C3) is different from that of the two EFRTs, but the entire wind speed isopleth inclines along the right side. Comparing the wind fields of the three cases at the same ambient wind speed, it can be seen that the area corresponding to a higher airflow speed becomes larger and the space of the blue backflow zone on the back becomes larger and more complex with an increase in the number of tanks. This is due to mutual blocking between tanks, which aggravates turbulence. In addition, due to interaction between the four large EFRTs, there is no longer a high-speed vortex over C3 on the leeward side.

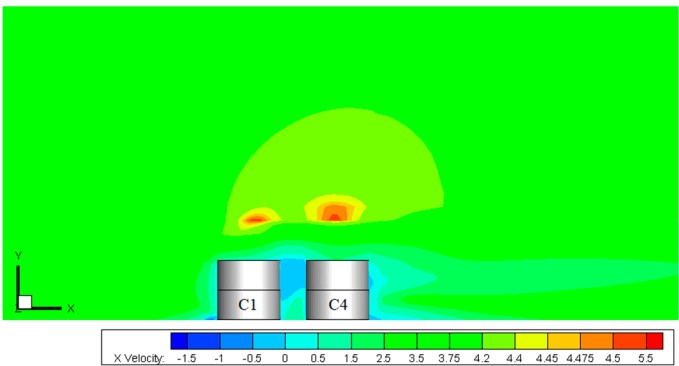

**Figure 10.** Wind speed cloud diagrams of C1 and C4 on the XY plane along the X axis (4 m/s).

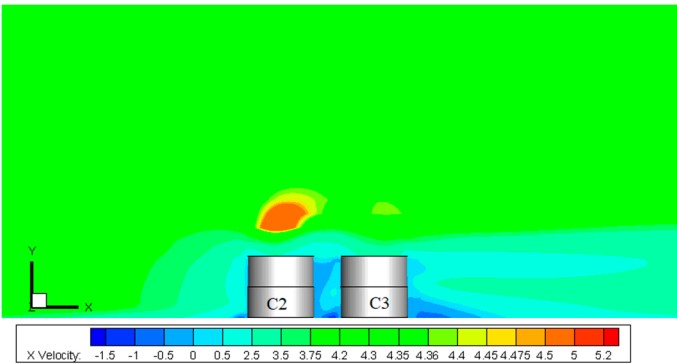

**Figure 11.** Wind speed cloud diagrams of C2 and C3 on the XY plane along the X axis (4 m/s).

### 4.2. Streamline Distribution Inside and Outside EFRTs

The streamline diagrams of gas movement in the single EFRT on the XZ plane at ambient wind speeds of 2 and 4 m/s are shown in Figure 12. Figure 13 illustrates the velocity vector diagrams of the single EFRT on the XY plane. The airflow follows a mirror distribution along the central axis of the floating deck. The vortex of the airflow is clockwise in the upper half and counter-clockwise in the lower half. Combining with the streamline diagrams of gas movement in the XY plane in Figure 13, the centre of the vortex is close to the middle of the floating deck. Comparing Figure 12a,b and Figure 13a,b, it can be inferred that the trend of gas movement in the single EFRT is almost constant.

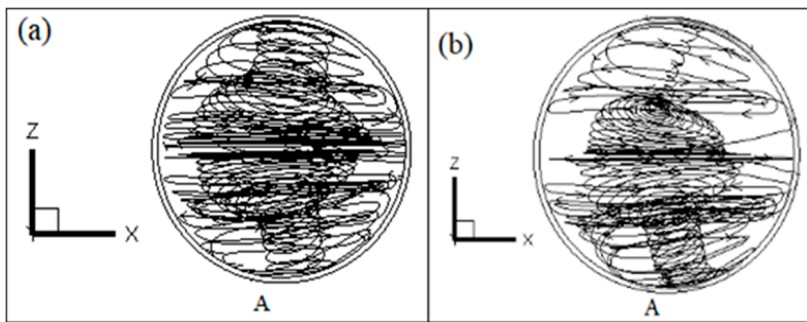

**Figure 12.** Flow diagrams of vapor movement in the single EFRT on the XZ plane at ambient wind speeds of (**a**) 2 and (**b**) 4 m/s.

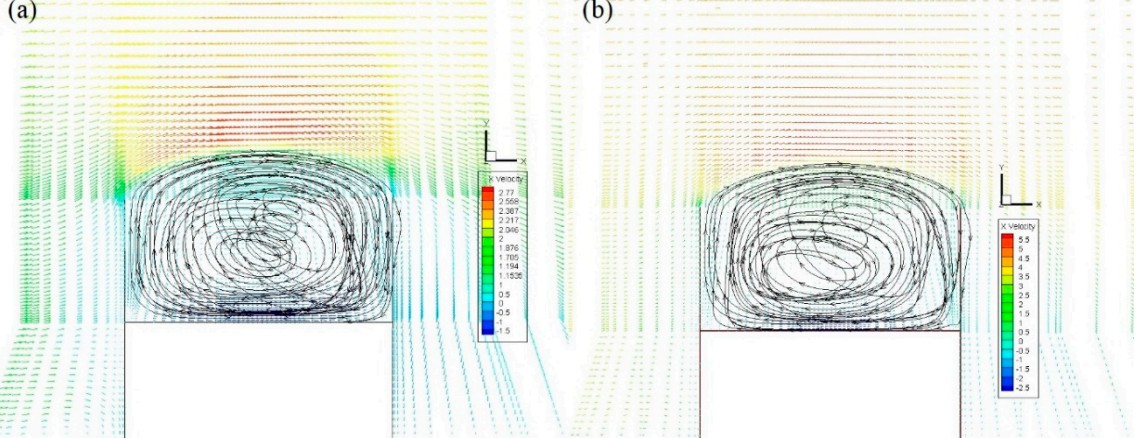

**Figure 13.** Flow diagrams of the vapor movement in the single EFRT on the XY plane at ambient wind speeds of (**a**) 2 and (**b**) 4 m/s.

The streamline diagrams of gas movement in the double EFRTs on the XZ plane at ambient wind speeds of 2 and 4 m/s are shown in Figures 14 and 15, respectively. Figure 16 shows the velocity-vector diagrams of the two EFRTs on the XY plane, in which the gas movement is more complicated. The front tank (B1) has two gas vortices that are similar to the single tank (A). Because of the blocking of B1 and disturbance in the airflow from B1, vortices in the rear tank (B2) are disturbed and no longer form recirculating vortices. According to Figures 14–16, gas movement in the single EFRT and two EFRTs is similar at ambient wind speeds of 2 and 4 m/s and hence we shall discuss the situation observed at 4 m/s later.

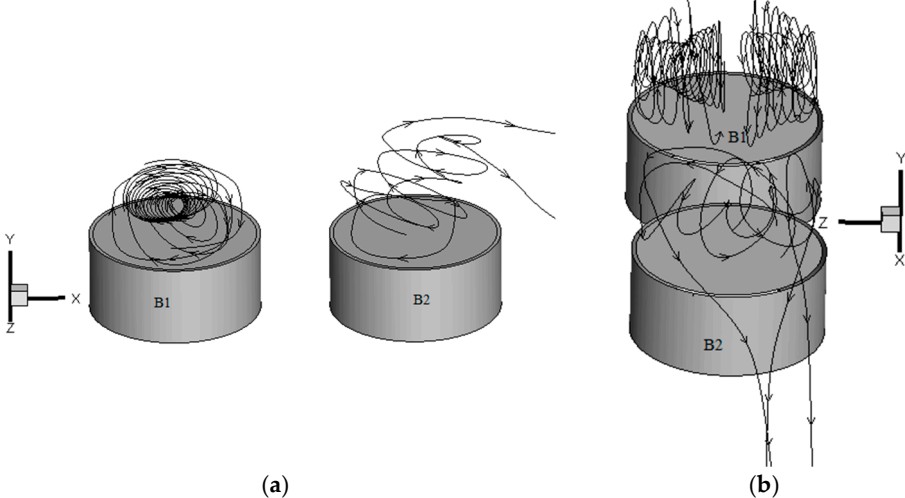

(**a**)  (**b**)

**Figure 14.** Flow diagrams of vapor movement in the two EFRTs at 2 m/s. (**a**) Positive angle and (**b**) side angle.

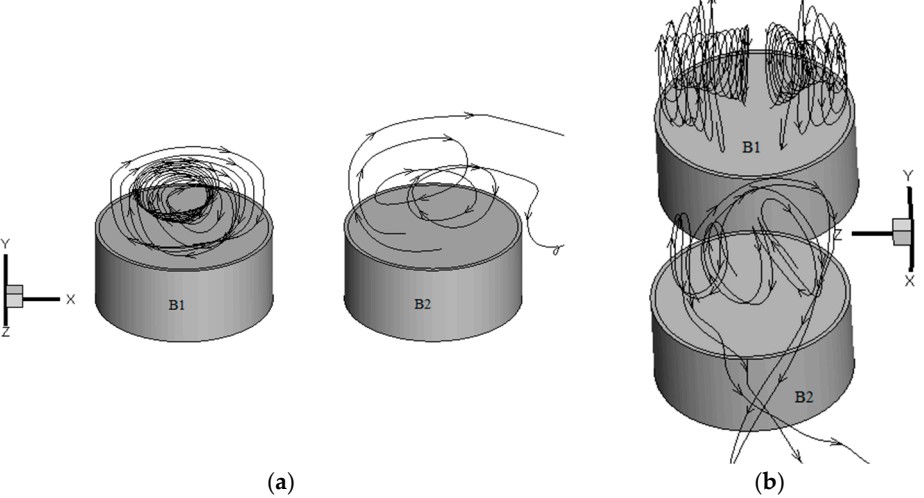

(**a**)  (**b**)

**Figure 15.** Flow diagrams of vapor movement in the two EFRTs at 4 m/s. (**a**) Positive angle and (**b**) side angle.

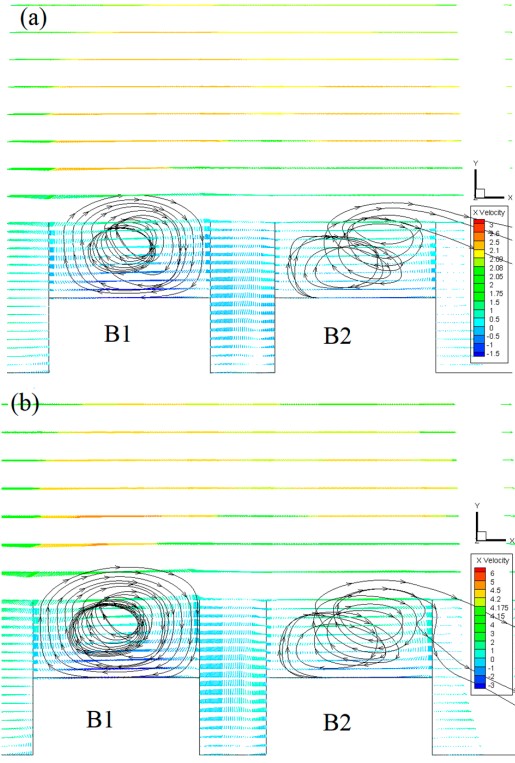

**Figure 16.** Flow diagrams of vapor movement in the two EFRTs on the XY plane at ambient wind speeds of (**a**) 2 and (**b**) 4 m/s.

The streamline diagrams of gas movement in the four large EFRTs on the XZ plane are shown in Figure 17. Here, the gas movement is highly complicated because apart from the interaction between the front and rear tanks, left and right EFRTs also exert some influence. The gas movement in C1 and C2 on the windward side is more regular and there are relatively complete airflow vortices in the tanks. In Figures 17 and 18, because of the effect of the Karman Vortex Street, airflow moves to the rear EFRTs periodically along a similar 'S' trajectory after bypassing the front EFRTs. Combining with the pressure cloud diagram in Figure 18, it can be seen that the pressure on the left side of C4 is higher than that observed for C3, which leads to a greater internal wind speed in C4 and large circular vortices. However, because the wind speed in C3 is too small to drive all the airflow in the tank, two symmetrical small circular vortices are formed on the left side.

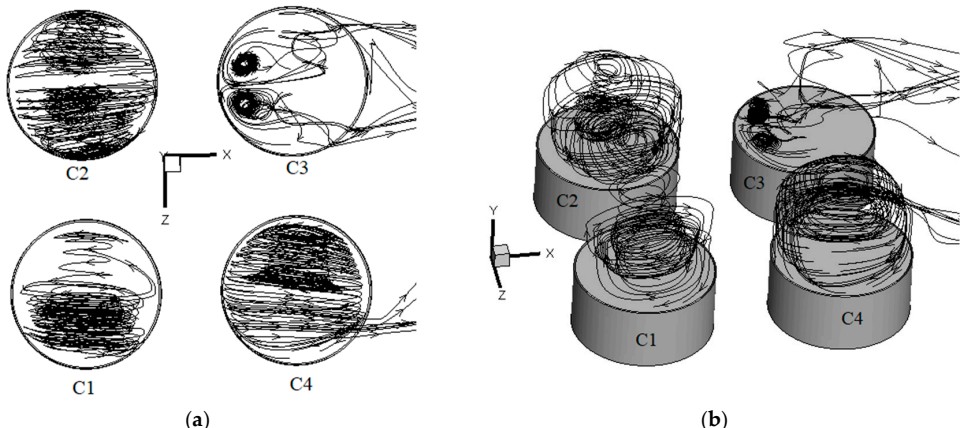

**Figure 17.** Flow diagrams of vapor movement in the four large EFRTs at an ambient wind speed of 4 m/s. (**a**) Top view and (**b**) main view.

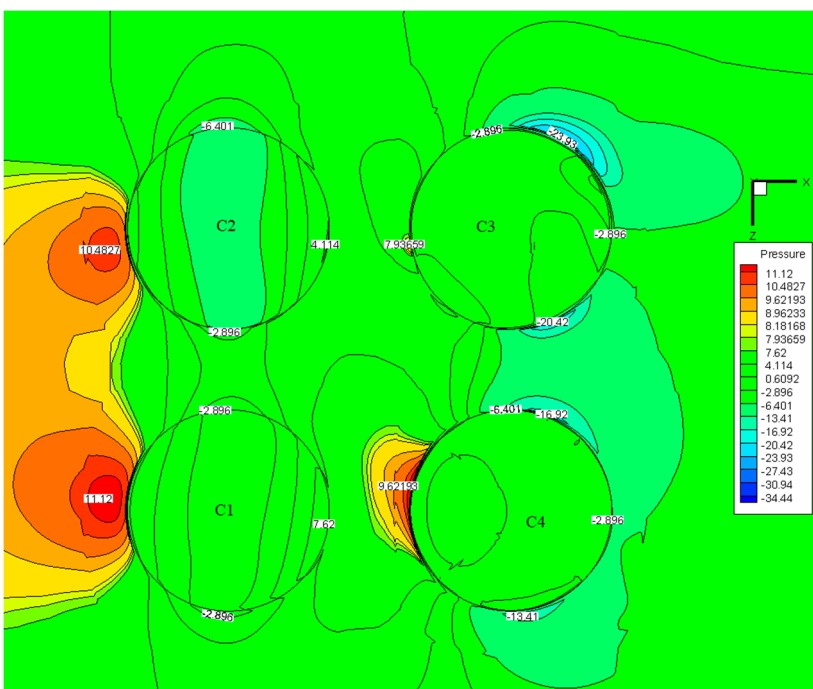

**Figure 18.** Pressure cloud diagram of the four large EFRTs at a height equal to the tank top on the XZ plane at an ambient wind speed of 4 m/s.

The flow diagrams of vapor movement outside different EFRTs (single, two, and four) at an ambient wind speed of 4 m/s are shown in Figures 19–21, respectively. Similar to the case of gas movement in EFRTs, as the number of tanks increases, the interaction between EFRTs increases and the trajectory of airflow becomes more complex. Vortices are formed but the vortex area of the two EFRTs and four large EFRTs is larger than that of the single EFRT. For the coupled and four large EFRTs, because the rear tanks block the backward movement of airflow, a backflow is also formed between them, resulting in vortices. This area also experiences vapor superposition and hence is a key monitoring area.

In addition, comparing Figures 19b, 20b and 21b, it can be seen that the vortex at the rear of the single tank (A) is stacked on one side but the vortex currents behind tanks B1, C1, and C2 accumulate symmetrically. Meanwhile, gas streamlines from the front tank (B1) and rear tank (B2) intersect behind the rear tank (B2). Similarly, gas streamlines from the front tank (C2) and rear tank (C4) intersect behind the rear tank (C4). Gas streamlines from the front tank (C2) and rear tank (C3) intersect behind the rear tank (C3). The area in which gas intersection occurs will also experience vapor superposition.

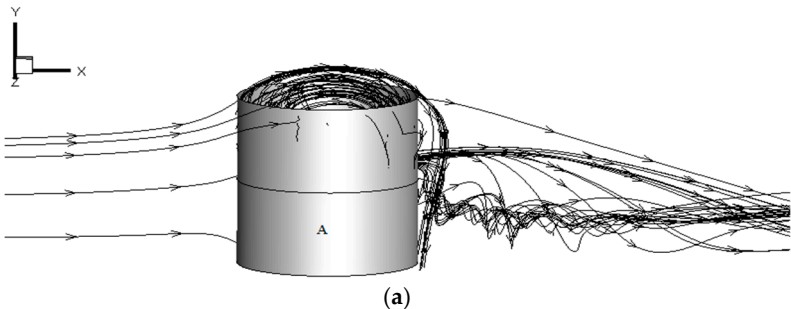

(**a**)

**Figure 19.** *Cont.*

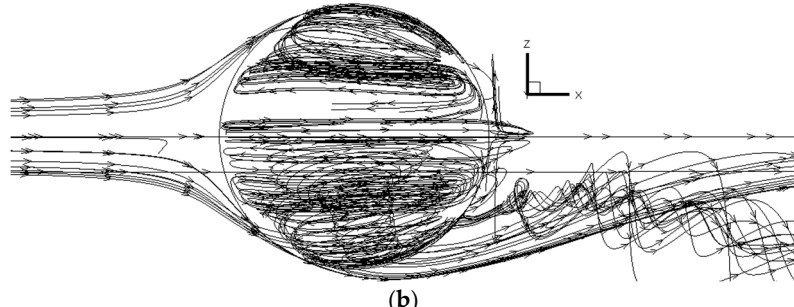

(**b**)

**Figure 19.** Flow diagrams of vapor movement outside the single EFRT at an ambient wind speed of 4 m/s. (**a**) Main view and (**b**) top view.

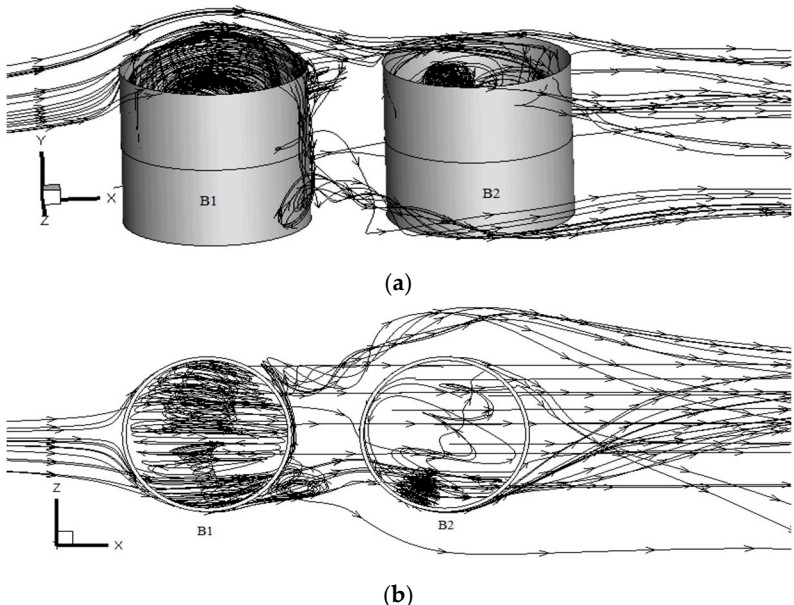

**Figure 20.** Flow diagrams of vapor movement outside the two EFRTs at an ambient wind speed of 4 m/s. (**a**) Main view and (**b**) top view.

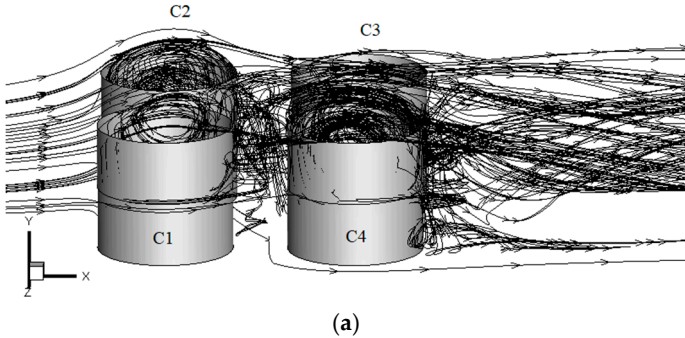

(**a**)

**Figure 21.** *Cont.*

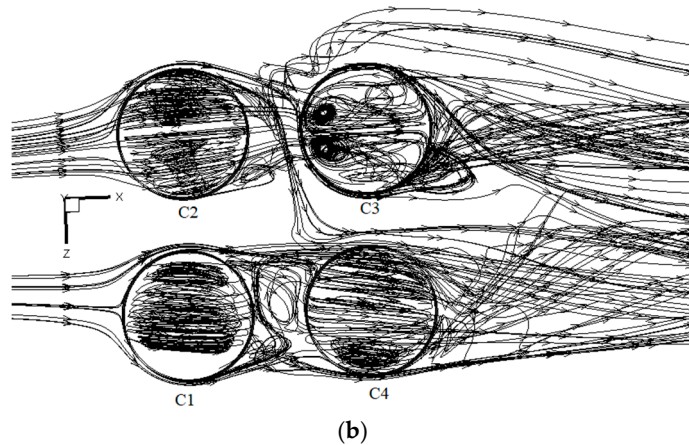

(**b**)

**Figure 21.** Flow diagrams of vapor movement outside the four large EFRTs at an ambient wind speed of 4 m/s. (**a**) Main view and (**b**) top view.

### 4.3. Concentration Distribution for Various EFRTs

Vapor mass-fraction distribution cloud diagrams corresponding to single EFRT on the XY plane are shown in Figure 22. It can be seen that when the floating deck rim leaks, vapors are mainly located near the rim and upper part of the floating deck surface, leading to vapor concentration and potential safety hazards. Combining these inferences with Figure 13, it can be stated that because the gas in the tank rotates upwards in a large vortex, vapor accumulates at the centre of the vortex and upper part of the gap between the floating deck and tank wall. The main reason is that airflow in the tank rotates clockwise and wind speed is very low near the floating deck surface, owing to which the vapor can easily accumulate. Comparing Figure 22a,b, it can be seen that when the ambient wind speed increases, turbulence in the airflow in the tank increases when the floating deck rim leaks and subsequently, vapor concentration above the floating deck increases.

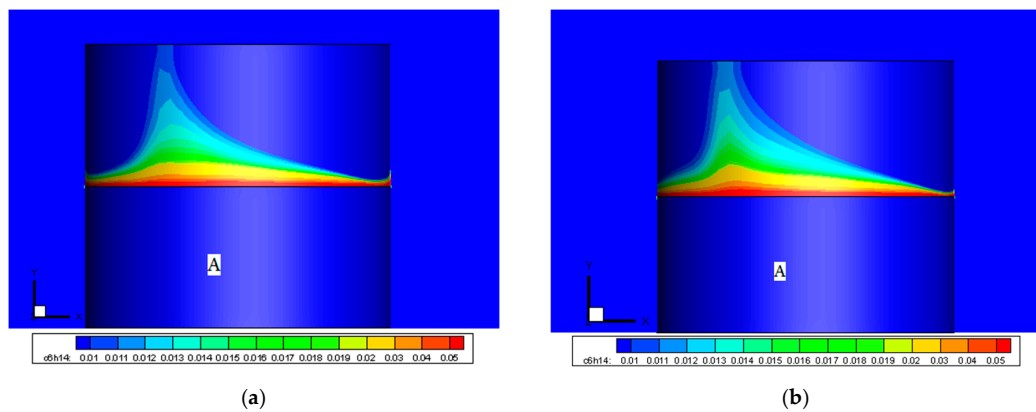

(**a**) (**b**)

**Figure 22.** Vapor concentration-distribution cloud diagrams in the single EFRT on the XY plane at ambient wind speeds of (**a**) 2 and (**b**) 4 m/s.

Vapor concentration-distribution cloud diagrams in the two EFRTs above the floating deck and at the tank wall are shown in Figures 23 and 24, respectively. The highest vapor concentration is found at the rim gap of the floating deck. Because the front tank (B1) blocks the rear tank (B2) and some ambient wind bypasses B1 and enters into B2 directly from the rear of B2 resulting in right-to-left vortices, it leads to vapor accumulation on the left side of the rim gap. At a low ambient wind speed of 2 m/s, the leaked vapor from B1 cannot be blown out of the tank and hence vapor concentration in this tank is very high. Due to the blocking of the front tank (B1), airflow speed in the rear tank (B2) is close to the ambient wind speed and hence vapor concentration in B2 is smaller than that at 4 m/s. Combining Figures 15 and 20, it

can be seen that vapor in the front tank (B1) moves upwards along the windward side of the tank wall due to the front airflow vortices and hence vapor concentration is higher on the left side than on the right side. After vapor in the rear tank (B2) leaks out from the gap of the floating deck, it mainly moves towards the tank top along the windward side of the tank wall and eventually accumulates in the left half of B2.

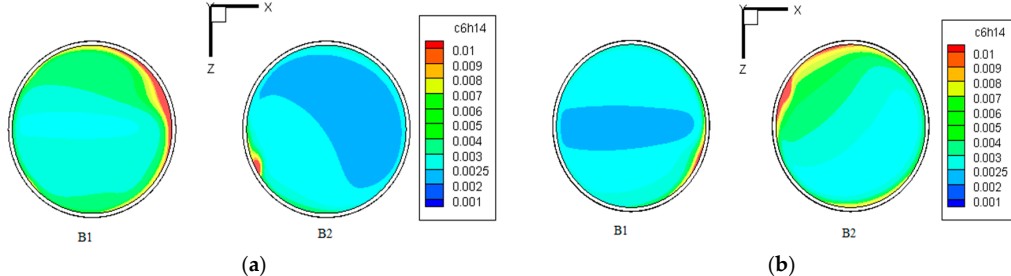

(**a**)  (**b**)

**Figure 23.** Vapor concentration-distribution cloud diagrams of the two EFRTs on the XZ plane at ambient wind speeds of (**a**) 2 and (**b**) 4 m/s.

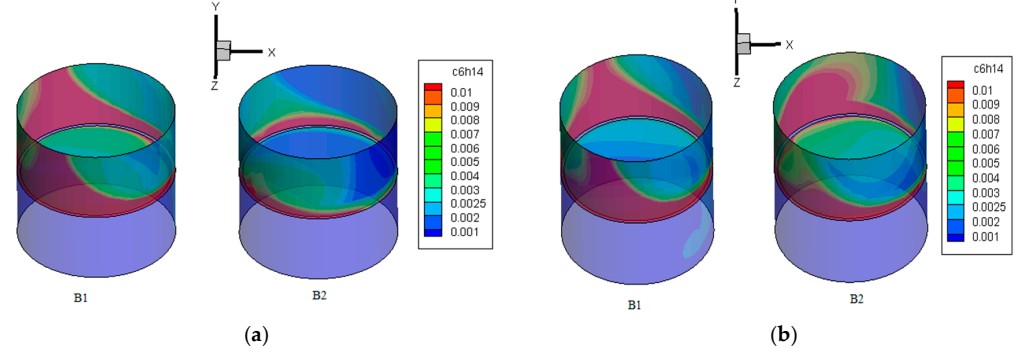

(**a**)  (**b**)

**Figure 24.** Vapor concentration-distribution cloud diagrams of the two EFRTs at ambient wind speeds of (**a**) 2 and (**b**) 4 m/s.

Vapor concentration-distribution cloud diagrams above the floating deck in the four large EFRTs on the XY plane at an ambient wind speed of 4 m/s are shown in Figure 25. It can be observed that vapor concentration in C2 is the lowest. This is because there are high-speed vortices above C2, which drive airflow in C2 in a clockwise manner and remove the leaked vapor. Figure 26 shows the vapor-concentration cloud diagram of the four large EFRTs near the ground on the XZ plane and Figure 27 depicts the vapor-concentration cloud diagram of the four large EFRTs at a height equal to the tank top on the XZ plane. Vapor concentration is relatively higher between C2 and C3 and after C3 and C4. From Figure 27, it can be inferred that vapor concentration in C4 is the highest, followed by C3; further, vapor concentration in C3 and C4 is larger than that in C1 and C2. In addition, according to Figures 21, 26 and 27, vapor superposition occurs behind C3 and C4 after leakage. Therefore, EFRTs in the downwind direction and the rear of these EFRTs should be considered as key areas for monitoring.

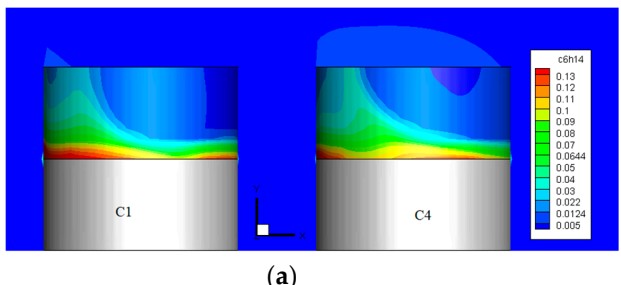

(**a**)

**Figure 25.** *Cont.*

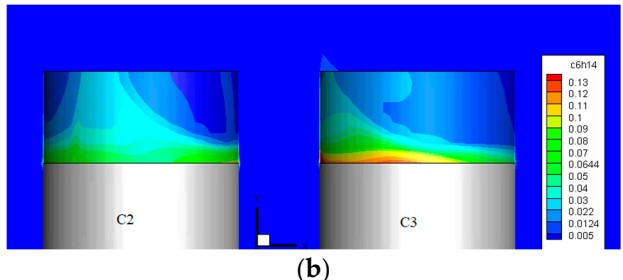

(**b**)

**Figure 25.** Vapor concentration-distribution cloud diagrams in the four large EFRTs on the XY plane of (a) C1 and C4, (b) C2 and C3.

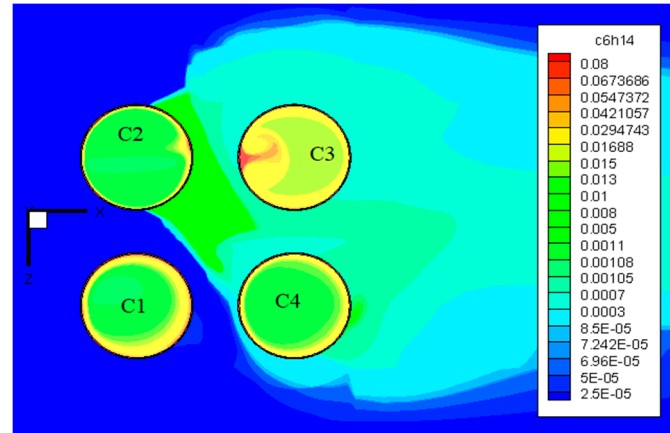

**Figure 26.** Vapor-concentration cloud diagram of the four large EFRTs near the ground on the XZ plane at 4 m/s.

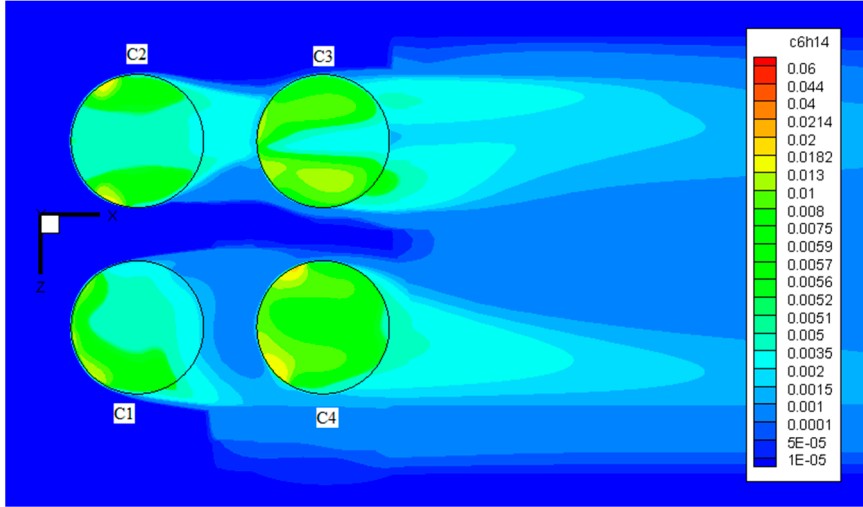

**Figure 27.** Vapor-concentration cloud diagram of the four large EFRTs at a height equal to that of the tank top on the XZ plane at 4 m/s.

## 5. Conclusions

In this study, we conducted numerical simulations and wind-tunnel experiments on vapor leakage and diffusion from a single EFRT and two EFRTs as well as numerical simulations on vapor leakage and diffusion from four large EFRTs. Based on wind-tunnel experiments, the physical model and numerical simulation model were verified. Furthermore, we discussed the distribution of wind speed and concentration fields in different types of EFRTs. Vapor diffusion after leakage from the rim gap of

the floating deck was studied and the superposition effect of the two tanks and four large tanks was investigated. The main features and conclusions of this work can be summarised as follows:

(1) A numerical simulation method for leakage in and diffusion from tank groups is proposed and verified by wind-tunnel experiments and it can be used to simulate leakage in and diffusion from tank groups of different numbers under different working conditions.

(2) For different EFRTs (single, two, and four), distributions on the windward side are similar. There is a large backflow area where the overall trend moves downwards on the leeward side. The two and four EFRTs also form gas vortices between the tanks and vapor tends to accumulate in them.

(3) At different ambient wind speeds, the interference between the two tanks is different. At 2 m/s, vapor concentration in the rear tank is smaller than that in the front tank. However, at 4 m/s, vapor concentration in the rear tank is higher than that in the front tank. Combining experimental and simulation results, when the ambient wind speed is greater than 2 m/s, vapor concentration in the leeward area of the rear tank is greater than that between the two tanks. It is suggested that more monitoring should be carried out at the bottom area of the rear tank and upper area on the left of the floating deck.

(4) The superposition effect becomes more obvious with an increase in the number of EFRTs. Vapor superposition occurs behind C3 and C4 after leakage from four large EFRTs. Therefore, EFRTs in the downwind direction and the area behind the EFRTs should be monitored frequently.

**Author Contributions:** Conceptualization, J.F., W.H.; software, J.F.; formal analysis, W.H., J.F., F.H., L.F. and G.Z.; investigation, F.H.; data curation, W.H., J.F., and G.Z.; writing—original draft preparation, J.F., W.H., F.H.; writing—review and editing, J.F., W.H., F.H., L.F. and G.Z.; supervision, W.H.; funding acquisition, W.H. All authors have read and agreed to the published version of the manuscript.

**Funding:** This research was funded by the National Natural Science Foundation of China (No. 51574044 and No. 51804045), the Key Research and Development Program of Jiangsu Province (Industry Foresight and Common Key Technology) (No. BE2018065), the Sci & Tech Program of Changzhou (No. CJ20180053) and Postgraduate Research & Practice Innovation Program of Jiangsu Province (NO. KYCX19_1786 and NO. SJCX19_0668).

**Conflicts of Interest:** The authors declare no conflict of interest.

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
