# Peer review of "Investigation of the Superposition Effect of Oil Vapor Leakage and Diffusion from External Floating-Roof Tanks Using CFD Numerical Simulations and Wind-Tunnel Experiments"

_processes, doi:10.3390/pr8030299_

Round 1

Reviewer 1 Report

I have reviewed the paper "Investigation of the superposition effect of oil vapour leakage and diffusion from external floating-roof tanks using CFD numerical simulations and wind-tunnel experiments". The paper deals with a numerical simulation of the airflow around the external floating-roof tanks and it studies how the oil leaks from the tank. The numerical results are verified by the wind-tunnel experiment.

I found the paper interesting and I believe that it is worth to be published in Processes after several comments are taken into account.

1) The authors are using the compressible Navier-Stokes model coupled with the evolution equation for the temperature, transport equation for the concentration and k-epsilon turbulence model. However, the description of the model given by equations (1)-(6) is unclear and contains several mistakes:
1a) It is a compressible model, but it is not written what pressure p is. How is it related to density and temperature (what state equation is used)?
1b) Momentum equation (2) contains only shear viscosity (which should be multiplied by symmetric part of the velocity gradient - this is a mistake), but no bulk part is present...
1c) Is density rho somehow related to the concentration omega?
1d) How does the solution of the k-epsilon turbulence model influence the solution of the momentum equation? It seems that it contains only a constant shear viscosity mu_t.
1e) The evolution equation for temperature (3) deserves some comments. It contains neither standard Newtonian dissipative heating nor thermal expansivity term.
1f) Can the authors provide the reference for the equation (3) or comment on the last term?

2) The complete specification of all boundary conditions is missing. For example, what BC is prescribed for the concentration omega?

3) What initial condition is used for the concentration omega?

4) How is the rim gap done? Is it done using the fine mesh?

5) The agreement between the experiment and the finite volume simulation is hard to believe. Is it possible to explain how the material parameters for the simulation were obtained and provide the values in the paper (they are not available in the paper)? I would understand if the simulation was a result of fitting the experimental data, but if it was a prediction using the material parameters from the literature, the agreement is extremely good. 

6) How the results would change if you did not use the k-epsilon model?

Some typos:

l.193: gird -> mesh
l.193: total number of grids -> total number of cells
l.195: Fig.4.The -> Fig.4. The
l.235: The caption of Fig. 6 has overflowed to the other side.

Author Response

Dear Reviewer,

Reviewer 2 Report

The manuscript was pretty well written, with a lot of experiments data and CFD simulation results. I have two minor question:

The figure 4 and figure 5 are recommended to be removed. It is not necessary to introduce the meshing and grid into this detailed. The context description is sufficient. How is the CFD model validated? It seems the experimental data and CFD results are discussed separately. This is still fin. But the CFD model need to be validated to show its effectivity.

Author Response

Dear Reviewer,

Reviewer 3 Report

The manuscript investigated the leakage and diffusion from the external floating-roof oil tanks. Both the wind tunnel experiments and simulation were conducted with detailed explanation and discussion.I hardly have any issue as the manuscript is very systematic and well presented, which deserves publication. 

Author Response

Dear Reviewer,

Round 2

Reviewer 1 Report

Dear authors,

I like the overall merit of your paper, but your answers to my questions are not satisfactory. Mainly, my questions concerning the mathematical model.

Ad 1a) For the compressible model you need to specify the equation of state, which is a relation between the pressure, density, and temperature. I am sure, that in Fluent you have to choose it. Add the equation of state that you are using to the paper.

Ad 1b) Your momentum equation does not include the term grad div v that is responsible for the volumetric changes of the air, which is standard in a compressible model. Can you justify such simplification?

Ad 1c) Can you provide the information how the density rho depends on concentration omega?

Ad 1d) Surely the solution of k-epsilon model has to influence the solution of the momentum equation, otherwise, there would be no need to use the k-epsilon model. My question directed to the fact that possibly the k-epsilon model influences the momentum equation through the eddy viscosity mu_t, which is not a kinetic viscosity. This should be incorporated in the paper.

Ad 1ef) The evolution equation for temperature (3) is missing some classical terms. If you can not justify the form of the presented equation, can you at least provide its source?

Ad 5) Can you provide all material parameters used in the governing equations?

Please take your time and answer carefully all the questions.

Author Response

Dear Reviewer,

Reviewer 2 Report

Significant improvement has been made to the manuscript. It is recommended to be accepted for publication. 

Author Response

Dear Reviewer,

Thank you for your review report and very good evaluation to our manuscript. Even so, we are also trying our best to improve the quality of our manuscript and the corrections in the revised manuscript are marked by using the "Track Changes". Thanks again for your review.

Round 3

Reviewer 1 Report

Thank you for your answers. Just modify the continuity equation to div v = 0. After that, I agree to accept the paper.

Author Response

Dear Reviewer,

Thank you for your many times review and guidance. Thank you for your suggestion and we have modified the continuity equation and the corrections in the revised manuscript are marked by using the "Track Changes". Thanks again for your review.